# Targeting NLRP3 Inflammasome Activation in Severe Asthma

**DOI:** 10.3390/jcm8101615

**Published:** 2019-10-04

**Authors:** Efthymia Theofani, Maria Semitekolou, Ioannis Morianos, Konstantinos Samitas, Georgina Xanthou

**Affiliations:** 1Cellular Immunology Laboratory, Center for Basic Research, Biomedical Research Foundation of the Academy of Athens, 11527 Athens, Greece; 27th Respiratory Clinic and Asthma Center, ‘Sotiria’ Athens Chest Hospital, 11527 Athens, Greece

**Keywords:** severe asthma, innate immunity, immune regulation, NLRP3, IL-1β, allergic airway inflammation

## Abstract

Severe asthma (SA) is a chronic lung disease characterized by recurring symptoms of reversible airflow obstruction, airway hyper-responsiveness (AHR), and inflammation that is resistant to currently employed treatments. The nucleotide-binding oligomerization domain-like Receptor Family Pyrin Domain Containing 3 (NLRP3) inflammasome is an intracellular sensor that detects microbial motifs and endogenous danger signals and represents a key component of innate immune responses in the airways. Assembly of the NLRP3 inflammasome leads to caspase 1-dependent release of the pro-inflammatory cytokines IL-1β and IL-18 as well as pyroptosis. Accumulating evidence proposes that NLRP3 activation is critically involved in asthma pathogenesis. In fact, although NLRP3 facilitates the clearance of pathogens in the airways, persistent NLRP3 activation by inhaled irritants and/or innocuous environmental allergens can lead to overt pulmonary inflammation and exacerbation of asthma manifestations. Notably, administration of NLRP3 inhibitors in asthma models restrains AHR and pulmonary inflammation. Here, we provide an overview of the pathophysiology of SA, present molecular mechanisms underlying aberrant inflammatory responses in the airways, summarize recent studies pertinent to the biology and functions of NLRP3, and discuss the role of NLRP3 in the pathogenesis of asthma. Finally, we contemplate the potential of targeting NLRP3 as a novel therapeutic approach for the management of SA.

## 1. Introduction

Asthma represents a serious global health problem that affects 1%–18% of the population of all age groups. Its prevalence has increased in the last decades, especially among children [1]. Asthma is characterized by variable symptoms of wheezing, dyspnea, chest tightness, coughing, and reversible airflow obstruction, and is usually associated with airway hyperresponsiveness (AHR) to innocuous environmental allergens and chronic airway inflammation. Factors, such as allergen or irritant exposure, respiratory infections, exercise, climate changes, and stress, are responsible for the disparities and severity of asthma symptoms [1]. Asthma has been long considered as a heterogeneous chronic lung disease that encompasses multiple groups of patients characterized by varying features or phenotypes [2,3]. A small percentage of asthmatics exhibit severe disease exacerbations despite the fact that they are already under treatment with high doses of inhaled and/or systemic corticosteroids [2,3]. These patients suffer from severe asthma (SA) that is poorly controlled and, in some cases, life-threatening [4,5]. Although patients with SA comprise a small percentage of the total asthma population (5%–10%), they denote 50% of total healthcare costs, rendering SA a substantial health and socioeconomic burden [6,7]. SA is characterized by marked thickening and structural changes of the airway wall, excessive airway narrowing, and fixed airflow obstruction [6,7]. An in-depth understanding of the heterogeneity of SA and the immunological mechanisms underlying its pathophysiology is critical for the identification of novel biomarkers and molecular pathways that can be targeted in novel treatment modalities.

In the lung, innate immune responses provide the first line of defense against environmental signals, including pathogens, allergens, and other irritants, and act through downstream signaling by numerous extracellular and intracellular receptors, termed pattern recognition receptors (PRRs) [8,9,10]. NOD-like Receptor Family Pyrin Domain Containing 3 (NLRP3) is an intracellular PRR that detects microbial motifs, endogenous danger signals, and environmental irritants, and induces the formation and activation of the NLRP3 inflammasome. Although the NLRP3 inflammasome is essential for providing protective immunity, overactivation of inflammasome-mediated responses can cause excessive inflammation, tissue damage, and lead to chronic inflammatory diseases, including asthma [10,11]. In this review, we describe the immunological mechanisms underlying aberrant inflammatory responses in the airways and their link to SA pathogenesis. We also present the biology and functions of NLRP3 and discuss its role in the initiation and propagation of SA features. Finally, we present recent findings pertinent to targeting NLRP3 functions as a novel therapeutic approach for the control of inflammatory responses in the airways.

## 2. Severe Asthma Pathogenesis

### 2.1. Type 2 Asthma

To address SA complexity, the concept of asthma endotyping has emerged [12,13,14]. Depending on the type of immune cell responses implicated in disease pathogenesis, asthma endotypes are categorized as (a) type 2 asthma, characterized predominantly by T helper type 2 (Th2) cell-mediated inflammation and (b) nontype 2 asthma, associated with Th1 and/or Th17 cell inflammation [15,16,17].

Upon allergen exposure, dendritic cells (DCs) in the lung mucosa take up allergens, reach the mediastinal lymph nodes, and present allergen components to naive T cells in the context of major histocompatibility complex class II [18]. Allergens with proteolytic activity, such as those derived from house dust mites (HDM), pollen grains, fungi, and occupational sensitizers activate protease activated receptors expressed on DCs, disrupt epithelial tight junctions, and initiate inflammatory responses [18]. Moreover, certain allergens and airborne particulates contain microbial components which interact with PRRs, including Toll-like receptors (TLRs) and NOD-like receptors (NLRs), on DCs and airway epithelial cells, and serve as “danger signals” for the host immune response [18]. Upon interaction with allergen-loaded DCs, naive Th cells differentiate into Th1, Th2, Th9, or Th17 cells, depending on the type and dose of allergen and the local cytokine milieu [18]. Allergen-specific Th2 cells, generated in the presence of type 2 cytokines, migrate into the airways wherein upon allergen re-exposure, secrete cytokines and promote mucus secretion, subepithelial fibrosis, bronchial remodeling, and AHR [19]. The production of Th2 cytokines also leads to the recruitment of innate effector cells, including mast cells, basophils, and eosinophils, as well as to isotype switching of B cell-secreted IgG to allergen-specific IgE [19]. Additionally, Th9 cells, a recently identified Th cell subset characterized by high levels of IL-9, exacerbate allergic airway inflammation (AAI), predominantly through activation of mast cell functions [20,21]. More specifically, experimental studies have shown that IL-9 production by Th9 cells and by type 2 innate lymphoid cells (ILC2s) enhances the production of IL-2 by mast cells, leading to further expansion of ILC2s, which activate Th9 cells, in a positive feedback loop [22]. Of clinical relevance, increased numbers of Th9 cells were demonstrated in peripheral blood mononuclear cells (PBMCs) isolated from HDM or pollen allergic subjects and correlated with IgE levels [23]. Moreover, elevated IL-9-secreting T lymphocytes were observed in the bronchoalveolar lavage (BAL) of asthmatics [24].

Apart from DCs, the asthmatic airway epithelium represents a major source of cytokines termed “alarmins”, such as, IL-25, IL-33, and thymic stromal lymphopoietin (TSLP), and chemokines, including RANTES (Regulated on Activation, Normal T cell Expressed and Secrete or CCL5), TARC (Thymus- and Activation-Regulated Chemokine or CCL17), eotaxins (CCL11, CCL24 and CCL26), and MCP-3 (Monocyte Chemotactic Protein-3 or CCL7) that trigger Th2 cell polarization upon exposure to allergens, pollutants, viral, fungal, and bacterial components [18]. As mentioned above, recent studies have highlighted a key role for ILC2s in asthma immunopathogenesis [25]. ILC2s are activated in response to TSLP, IL-25, and IL-33 signaling [25], and produce IL-5, IL-13, and prostaglandin [26], further propagating Th2-cell mediated responses in the airways.

Type 2 asthma is characterized by any combination of the following processes: eosinophilia in the sputum or blood, atopy and a high level of fractional exhaled nitric oxide (FeNO) [16]. Several biomarkers of type 2 inflammation, such as FeNO, serum IgE, blood or sputum eosinophils, and serum periostin distinguish type 2-high and type 2-low asthma phenotypes and also predict the responsiveness to type 2 cytokine-targeted therapy [27]. Eosinophils play a crucial role in the initiation and propagation of inflammatory responses in asthma [16]. Asthmatic patients with increased numbers of eosinophils in the periphery suffer from more severe disease exacerbations and poorer disease control [16]. FeNO represents an indicator of IL-13-mediated and corticosteroid-responsive airway inflammation, as the presence of IL-13 activates inducible nitric oxide synthase (iNOS), leading to increased FeNO production in the airways [16]. Approximately 70% of patients with asthma have an allergic phenotype, characterized by allergen-specific IgE and elevated total IgE levels [16]. Periostin, an extracellular matrix protein, induced by IL-4 and IL-13 signaling, is secreted by bronchial epithelial cells and represents another important biomarker for severe eosinophilic type 2 asthma [28]. Periostin is involved in airway remodeling, sub-epithelial fibrosis, eosinophil recruitment, and mucus production. Notably, a high serum concentration of periostin denotes one of the most significant indicators of eosinophilic inflammation in asthma [29]. 

Clustering studies revealed that one of the major reasons SA patients remain unresponsive to corticosteroid treatment is that apart from Th2 inflammation, other mediators are also implicated in disease pathogenesis [12,30]. Indeed, enhanced Interferon gamma *IFNG* expression was detected in BAL cells, accompanied by increased secretion in SA patients compared to mild-to-moderate asthmatics (MMA) [31]. The same study also showed elevated percentages of CD4^+^IFN-γ^+^ T cells in the BAL [31]. In line with above, increased IFN-γ mRNA levels were observed in lung tissue and sputum of SA patients [32,33]. It must be asked what triggers these IFN-γ-mediated responses in the airways of SA patients? It is known that persistent viral (especially rhinoviruses) and bacterial (*Chlamydia pneumoniae*, *Streptococcus pneumoniae*, *Mycoplasma pneumoniae*, *Haemophilus influenzae*, *Moraxella catarrhalis* and *Staphylococcus aureus*) infections augment IFN-γ secretion in SA patients, highlighting a key role for respiratory tract infections in disease severity and asthma exacerbations [34].

### 2.2. Nontype 2 Asthma

The pathophysiology of nontype 2 asthma remains less well characterized. Nontype 2 asthma is denoted by the absence of type 2 biomarkers, such as eosinophils and IgE, and the predominance of Th17 cells and neutrophils in the airways [35]. In fact, growing evidence has highlighted a key role for IL-17 in the pathogenesis of nontype 2 asthma [35]. Tissue-infiltrating CD4^+^ T cells isolated from bronchial biopsies of SA patients produce copious amounts of IL-17 and IL-22 upon ex vivo polyclonal stimulation [36,37]. Another study revealed that in vitro administration of recombinant human IL-17 enhanced the secretion of IL-8 by human bronchial epithelial cells (HBECs) and venous endothelial cells [38]. Moreover, conditioned medium from IL-17-treated HBECs increased neutrophil migration in vitro [38]. Increased IL-17 mRNA levels are also observed in the sputum of SA patients [39]. In addition, IL-17 levels in the peripheral blood (PB) of SA patients positively correlate with disease severity [40]. A very interesting recent study revealed that an IL-4Rα polymorphism associated with SA leads to the conversion of regulatory T cells (Tregs) to Th17 cells in vitro [41]. In brief, the authors isolated from the PB of asthmatics and healthy controls that were either wild type (WT) (IL4RQ576/Q576), heterozygous (IL4RQ576/R576), or homozygous (IL4RR576/R576) for the mutated allele, naive T cells, and differentiated them into Treg cells. Interestingly, naive CD4^+^ T cells from asthmatics bearing the IL4RR576 mutation exhibited defective induction of Treg cells and were skewed towards a Th17-like phenotype, exemplified by increased secretion of IL-17 [41]. Still, targeting IL-17 has failed to improve disease symptoms in SA patients as opposed to anti-type 2 cytokine therapy, suggesting that other mechanisms compensate for its pathogenic effects or that targeting pathogenic Th17 cells specifically would be more appropriate [42,43].

To date, biomarkers of type 2-low or neutrophilic asthma have not been identified. Several studies have shown that there is no inverse correlation between sputum eosinophil and neutrophil numbers in SA patients and that eosinophils are present in excess, additionally to neutrophilic accumulation, in the airways [44]. Moreover, although measuring eosinophil numbers in the periphery correlates with the percentages of eosinophils in induced sputum, blood neutrophil parameters represent poor surrogates for the proportion of neutrophils in the sputum [45,46]. Recent studies showed that the levels of the chitinase-like protein YKL-40 in the blood of SA patients could potentially be used as a marker for nontype 2 neutrophilic asthma [47]. In fact, combining the measurement of YKL-40 with other clinical parameters of the disease may provide a more reliable strategy for defining nontype 2 asthma. Tumor necrosis factor (TNF)-α has been also shown to have a critical role in nontype 2 asthma through acting on smooth muscle cells or by modifying the release of the cysteinyl leukotrienes, LTC4 and LTD4 [48]. TNF-α is increased in the BAL and bronchial biopsies in SA patients compared to MMAs [49]. Importantly, the administration of inhaled recombinant TNF-α to normal subjects led to the development of AHR and airway neutrophilia [50,51]. Subsequent clinical trials utilizing antiTNF-α therapeutic administration contributed to the elucidation of the role of this cytokine in vivo [52]. Indeed, improvement in patient quality of life, lung function, and a reduction in AHR and exacerbation frequency was observed in patients treated with antiTNF-α [52]. Still, considering the significant heterogeneity observed in SA patients, benefit from antiTNF-α therapy is likely to occur only in a small group of patients.

Apart from type 2 and nontype 2, asthma can be categorized into four endotypes (eosinophilic, neutrophilic, mixed granulocytic, and paucigranulocytic) based on the type of airway infiltrating immune cells [53]. Among these endotypes, paucigranulocytic asthma (PGA) presents no evidence of increased eosinophils or neutrophils in induced sputum, and instead is characterized by low-grade airway inflammation associated with airway smooth muscle (ASM) dysfunction, persistent airflow limitation, and AHR [54,55]. Molecules involved in oxidative stress, matrix metalloproteinases, neutrophil elastase, and galectin-3, commonly used for the discrimination between eosinophilic and neutrophlic asthma, also remain unaltered in patients with PGA [56,57,58,59]. Moreover, patients with PGA display lower FeNO levels compared to those with eosinophilic asthma [60]. Notably, Wang and colleagues documented that PGA was the most common endotype observed in children with stable asthma [61]. Corticosteroids do not seem to exert beneficial effects in this cohort of asthmatics, irrespective of the dose used [62]. Considering that symptoms dominating in this cohort are mostly due to ASM phenotypic changes and/or neuronal dysfunction, therapies directed toward ASM responses, including mast cell-targeted therapies and/or bronchial thermoplasty, might benefit PGA patients [63]. Although the precise mechanisms of action remain incompletely understood, bronchial thermoplasty is considered to diminish ASM mass through the delivery of localized thermal energy [63]. In addition, factors associated with dysregulated ASM functions, as well as mediators involved in the thickening of the subepithelial basement membrane, could be used as disease biomarkers and guide the design of effective therapeutic approaches for PGA [56].

In sharp contrast to adult asthma, wherein the type of airway inflammation has been extensively investigated due to the use of invasive and semi-invasive techniques, such as bronchial biopsies, BAL, and induced sputum, the majority of analyses in childhood asthma have been performed only in severe forms of disease [64,65,66,67]. The most common endotype of childhood asthma is the eosinophilic, characterized by unrestrained symptoms, atopy, impaired lung function, enhanced AHR, increased numbers of exacerbations, and steroid responsiveness [68]. The percentages of eosinophils in induced sputum significantly vary over time in children with SA, and these variations are not associated with asthma therapy [69]. Importantly, persistent airway eosinophilia has been detected in a small cohort of children with SA, even after a high dose of systemic corticosteroids [70]. Regarding neutrophilic pediatric asthma, the functional role of airway neutrophils in asthma pathophysiology remains elusive. Infiltration of neutrophils in the airways in pediatric SA represents a feature of airway inflammation at all ages and is triggered by viral and/or bacterial infections, exacerbating asthma symptoms [71]. Increased numbers of intraepithelial neutrophils, along with enhanced epithelial and submucosal expression of IL-17R, have been observed in lung biopsies of a subgroup of children with therapy-resistant SA. These findings correlate with improved lung function suggesting a potential beneficial rather than adverse role for neutrophils in pediatric SA pathophysiology [72]. Other studies show that neutrophils coexist with eosinophils in the airways of a group of children with SA, highlighting the complexity of defining the distinct endotypes in children [73]. Overall, it becomes evident that a more detailed and careful assessment of all the inflammatory endotypes is essential for the characterization and management of severe pediatric asthma. 

## 3. Targeted Therapies For Severe Asthma

In-depth investigation of the molecular and cellular mechanisms underlying SA pathophysiology has significantly contributed to the development of novel therapeutic strategies for disease management. Indeed, antibodies targeting factors involved in SA pathology are already being used as a first line medication. An excellent example is omalizumab, a monoclonal antibody directed against IgE that has become an established add-on therapy for patients with uncontrolled allergic asthma [74]. In addition, monoclonal antibodies against IL-5 (reslizumab, mepolizumab), IL-5R (benralizumab), and IL-4R (dupilumab) have been approved as add-on treatments for uncontrolled type 2 eosinophilic asthma [74]. Although these therapies have proven effective in certain asthma cohorts, some patients that suffer from severe allergic and/or eosinophilic asthma, as well as most patients with severe non type 2 asthma, experience weakly-regulated disease manifestations [74]. Notably, the reported adverse effects of these monoclonal antibody therapies should be also considered. For example, administration of omalizumab has been associated with an anaphylaxis rate of 0.09% that most frequently occurs within 2 hours after the first dose and 30 min after subsequent doses, necessitating the close monitoring of patients [75,76]. Moreover, a higher incidence of cardiovascular and cerebrovascular adverse effects (AEs) has been observed upon omalizumab administration [77]. Mepolizumab, the first anti-IL-5 monoclonal antibody approved for eosinophilc asthma, has been associated with headaches, injection site reactions, back pain, and fatigue [78]. The most common AEs of reslizumab, another FDA-approved anti-IL-5 antibody, are a 0.3% anaphylaxis rate, elevated serum creatinine kinase, and musculoskeletal and oropharyngeal pain [79]. Regarding benralizumab, an anti-IL-5R antibody that is currently undergoing phase 3 trials, there have been no documented AEs apart from nasopharyngitis and injection site reactions [80]. AEs in patients receiving dupilumab, a monoclonal antibody that targets the common receptor for IL-4 and IL-13, include nasopharyngitis, injection site reactions, and headaches [81]. Monoclonal antibodies targeting TSLP, IL-33 and its receptor ST2, the receptor for stem cell factors on mast cells, and a DNA enzyme directed at GATA3 are currently being evaluated for their efficacy in SA. It is worth mentioning that a number of antagonists directed against other potential targets, such as, IL-25, IL-6, TNF-like ligand 1A, CD6, and activated cell adhesion molecules are under consideration for future clinical trials [74]. Results from these clinical trials will be of great importance as they may introduce novel treatment modalities that will successfully replace the existing ones and lead to the efficient management of SA. 

Taken together, it becomes evident that as the airway lumen is continually exposed to external and endogenous stimuli, its ability to distinguish between innocuous environmental allergens and pathogenic agents is crucial for the maintenance of immune tolerance and lung homeostasis. In fact, the complex interactions between innate and adaptive immune responses in the lung micromilieu represent a major determinant of the development of tolerance or allergic inflammation. Innate immune responses have considerable bearing on ensuing adaptive responses, and if left uncontrolled, can lead to detrimental pathological consequences. Hence, delineation of the precise mechanisms involved in the regulation of innate immune reponses in the airways is essential for the design of more efficient treatment modalities for SA patients.

## 4. Inflammasomes: A Key Component of Innate Immunity

The innate immune system acts as the first line of defense during exposure to environmental pathogens. In the lung, innate immune responses act through downstream signaling by numerous extracellular and intracellular receptors termed pattern recognition receptors (PRRs). PRRs recognize pathogen-associated molecular patterns (PAMPs), such as lipopolysaccharide (LPS), bacterial and viral RNA, danger-associated molecular patterns (DAMPs) in damaged and/or dying cells, including reactive oxygen species (ROS), ATP and mitochondrial DNA, and homeostasis-altering molecular processes (HAMPs) that detect alterations in cell homeostasis and elicit inflammatory responses in the host [8,9]. PRRs are expressed on macrophages, monocytes, DCs, and on tissue-resident cells, including airway epithelial cells, and upon ligand binding, induce the secretion of inflammatory cytokines and chemokines [82,83]. PRRs consist of the Toll-like receptors (TLRs), the RIG-I-like receptors (RLRs), the nucleotide-binding oligomerization domain-like receptors (NLRs), the Scavenger receptors, the C-type lectin receptors (CLRs), and the absent-in-melanoma (AIM)-like receptors (ALRs). Among PPRs, TLRs and NLRs represent the most well-known and studied receptors [83,84]. TLRs are located at the cell membrane and the intracellular compartment, while NLRs are located solely in the cytosol [84]. Signaling through these receptors enables the innate immune system to monitor and respond to infectious agents and other damage-inducing stimuli, eliciting protective immunity.

NLRs consist of a conserved nucleotide binding and oligomerization domain (NACHT), a carboxy-terminal ligand-binding region, composed of leucine-rich repeats (LRRs), involved in ligand binding or activator sensing, and an amino-terminal effector domain required for protein–protein interactions [85,86,87]. The human NLR gene family consists of 22 members, classified into four subfamilies depending on their N-terminal regions: NLRA, NLRB, NLRC, and NLRP. The NLRA region contains an acidic transactivation domain, the NLRB a baculoviral inhibitory repeat-like domain, NLRC contains a caspase activation and recruitment domain (CARD), and NLRP contains a pyrin domain (PYD) [85,86,87]. The detection of PAMPs, DAMPs, and HAMPs by NLRs leads to the formation of a large multimolecular signaling platform called the inflammasome. Inflammasomes respond to a constellation of endogenous and pathogenic signals and are critical inducers of host defense. Five major inflammasomes have been identified so far: NLRP1, NLRC4, RIG-I, AIM2, and NLRP3. These consist of an active NLRP receptor, the inflammasome adaptor protein, Apoptosis-associated Speck-like protein Containing CARD (ASC), and caspase-1 [88]. NLRP6, NLRP7, NLRP12, and IFI16, can also form inflammasomes, but their composition remains unclear. Apart from the physiological role of inflammasomes in providing protective immunity, inflammasomes also regulate cell proliferation and tissue repair processes [11]. Still, overactivation of inflammasome-mediated responses can cause excessive inflammation, tissue damage, and lead to chronic inflammatory diseases and metabolic disorders.

## 5. Nlrp3 Biology and Functions 

NLRP3 is expressed in granulocytes, monocytes, DCs, T cells, and epithelial cells [89]. The NLPR3 inflammasome consists of the NLRP3 receptor, the adaptor protein ASC, also known as Pycard, and caspase-1 that acts as an effector protein [89]. The NLRP3 receptor is a tripartite protein that contains an amino-terminal PYD, a nucleotide-binding NACHT, and a carboxy-terminal LRR domain. ASC has two domains, an amino-terminal PYD and a carboxy-terminal CARD domain. NLRP3 activation leads to protein–protein interactions between the NLRP3 and ASC via PYD domains. This facilitates ASC polymerization to form long helical filaments that are condensed into an intracellular macromolecular aggregate, known as ASC speck [90]. Subsequently, the ASC CARD domain associates with the CARD domain of caspase-1, inducing caspase-1 self-cleavage and activation [91]. A serine–threonine kinase known as NIMA-related kinase 7 (NEK 7) binds to NLRP3 directly and oligomerizes with NLRP3 into a complex that is essential for ASC speck formation and caspase-1 activation [92]. Activated caspase-1 cleaves the inactive pro-IL-1β and pro-IL-18 forms into bioactive cytokines that activate downstream inflammatory pathways [93] (Figure 1). Recent studies have shown that ASC specks can be exocytosed and accumulated in the extracellular space, retaining their ability to produce IL-1β. Extracellular ASC specks can also be internalized by macrophages, further activating IL-1β production [94,95]. Moreover, ASC specks isolated from cells can induce the aggregation of other ASC specks located intracellularly, a feature shared with prionoid proteins. These findings suggest that extracellular ASC can propagate inflammatory responses even at distinct sites, promoting systemic inflammation. Of note, increased extracellular ASC specks were documented in bone marrow from patients with myelodysplastic syndrome [96]. Pertinent to lung diseases, increased extracellularly-assembled ASC specks were observed in the BAL of patients with chronic obstructive pulmonary disease and pneumonia compared to patients with pulmonary hypertension and healthy controls [94]. Considering the documented effects of extracellular ASC specks on the activation of distant cell populations, their presence in SA patients may have important implications, and warrants further investigation.

IL-1β secretion upon NLRP3 inflammasome activation initiates acute phase reactions, including the recruitment of inflammatory cells at the site of infection and expression of proinflammatory cytokines, such as, IL-6, TNF-α, and chemokines [97]. Briefly, active IL-1β binds to the extracellular domain of the IL-1 type 1 receptor (IL-1R) that recruits the second receptor chain, termed IL-1R accessory protein (IL-1RAcP) [98]. This leads to the activation of intracellular signaling molecules, such as the myeloid differentiation primary response 88 (MYD88), TNF receptor-associated factor 6 (TRAF6) and IL-1R-associated kinases (IRAK), which, in turn, activate the nuclear factor-κB (NF-κB) transcription factor, eliciting cytokine gene expression [99]. Active IL-1β exerts its functions through both autocrine and paracrine mechanisms, and therefore its regulation is under tight control to circumvent pathologic implications. IL-1β signaling is also involved in Th2 and Th17 cell differentiation and is implicated in the pathogenicity of allergic airway inflammation [100]. The biological role of IL-18 is different from that of IL-1β. IL-18 induces Th1 cell responses and the production of IFN-γ by CD4^+^ and CD8^+^ T cells, natural killer cells, and macrophages, and is essential for antiviral immunity. IL-18 also plays a role in Th2 cell differentiation and is involved in AHR in experimental models, as well as in asthmatic patients [101,102].

Apart from proinflammatory cytokine release, a key outcome of NLRP3 inflammasome activation is pyroptosis, a form of lytic cell death characterized by cell swelling, membrane rupture, and release of proinflammatory cellular contents [103,104]. The formation of plasma membrane pores during pyroptosis drives ion changes, inducing increased osmotic pressure, water influx, and cell swelling. Pyroptosis is triggered by caspase-1-driven cleavage of the pore-forming protein gasdermin D (GSDMD) that leads to the formation of the GSDM N-terminal fragment, that, in turn, introduces pore formation upon insertion into the plasma membrane, thus killing cells from within [103,104]. Pyroptosis is a highly inflammatory process that is accompanied by the release of IL-1β, IL-18, IL-1α, high mobility group box 1 protein, and lactate dehydrogenase to the extracellular milieu [103,105] (Figure 1). NLRP3 inflammasome activation occurs through three distinct molecular pathways: the canonical, the noncanonical, and the alternative pathways [11,86,89,92,105,106] (Table 1).

## 6. Activation of the Canonical Nlrp3 Pathway

Activation of the NLRP3 inflammasome through the canonical pathway requires 2 steps: a priming signal and a second activating signal (Figure 1). The priming signal is crucial for the transcription of the inflammasome components, *NLRP3*, *CASP1*, and the *IL1B* and *IL18* genes. The priming signal is usually a PAMP, such as, LPS, lipoproteins, carbohydrates, and flagellin [107]. IL-1β and TNF-α signaling can also induce NLRP3 priming, known as sterile priming. Gene transcription upon priming is mediated predominantly through NF-κB activation and nuclear translocation [108,109]. Interestingly, recent studies have discovered a nontranscriptional priming process that relies on post-translational modifications (PTMs), such as ubiquitylation, phosphorylation, and sumoylation of NLRP3 components [110].

A wide variety of secondary stimuli activate the NLRP3 inflammasome, including bacteria, viruses, environmental nanoparticles such as alum and silica, and endogenous molecules, including ATP, monosodium urate (MSU), and cholesterol crystals. The main mechanisms through which these PAMPs and DAMPs trigger NLRP3 inflammasome activation are associated with: a) changes in cytosolic levels of ions, such as, K^+^, Ca^+2^, and Cl^-^, b) lysosomal destabilization, c) ROS production, and d) mitochondrial dysfunction [92,111]. The second activation signal leads to the assembly of the NLRP3 complex, the activation of caspase-1, and the release of the mature forms of IL-1β and IL-18 (Figure 1).

Decreased extracellular K^+^ levels trigger NLRP3 activation, while high extracellular K^+^ concentrations block NLRP3 signaling [112]. For example, extracellular ATP, upon binding to its receptor, the purine-dependent phenoxin-1 channel P2X7, induces K^+^ efflux and initiates NLRP3 assembly and downstream signaling [113,114]. The bacterial toxin nigericin also promotes activation of NLRP3 by inducing K^+^ efflux in a pannexin-1-dependent manner [115]. In addition to ATP and pore-forming toxins, alum, silica, and calcium pyrophosphate crystals also induce K^+^ efflux. Moreover, the complement cascade component membrane attack complex (MAC) activates NLRP3 [116]. Apart from K^+^ efflux, mobilization of Ca^+2^ in the cytosol through the opening of plasma membrane channels or the release of endoplasmic reticulum (ER)-linked Ca^+2^ stores represents another upstream event in NLRP3 activation [117]. It should be emphasized that K^+^ efflux regulates Ca^+2^ flux and these two channels act cooperatively to activate NLRP3. In fact, NLRP3 activation induced by nigericin, alum, MSU crystals, and the MAC depends on Ca^+2^ flux along with K^+^ efflux [118]. Cl^-^ efflux through chloride intracellular channel proteins (CLICs) enhances NLRP3 activation via the polymerization of ASC [119]. Translocation of CLIC1, CLIC4, and CLIC5 to the plasma membrane depends on the release of mitochondrial ROS (mtROS), whereas Cl^-^ efflux occurs downstream of K^+^ efflux [119]. Lysosomal swelling and damage by phagocytosed but resistant to degradation crystals, such as silica, β-amyloid, liposomes, and asbestos, represents another mechanism of NLRP3 activation [120,121]. Briefly, the accumulation of crystals intracellulary destabilizes the lysophagosome and leads to the release of its components, including proteases, lipases, cathepsins, and Ca^+2^ in the cytosol, which, in turn, drive NLRP3 assembly and activation in a K^+^ efflux-dependent manner [120,121] (Figure 1).

Numerous sources of ROS, such as, NADPH-oxidases, xanthine oxidase, cytochrome P450, cyclooxygenases, and lipoxygenases, induce NLRP3 activation [122] (Figure 1). The production of mtROS and mitochondrial DNA (mtDNA) also activate NLRP3. In fact, chemical inhibitors preventing ROS production inhibit NLRP3 inflammasome activation in response to several activators. Furthermore, factors that cause mitochondrial dysfunction increase the oxidation of mtDNA, which activates NLRP3 inflammasome. Increased mtROS production oxidizes thioredoxin (TRX), leading to its dissociation from the thioredoxin (TRX)-interacting protein (TXNIP). The dissociated TXNIP directly binds to NLRP3, leading to its activation [123]. Mitochondria also act as docking sites for NLRP3 inflammasome assembly. For example, the mitochondrial antiviral signalling protein (MAVS), an adaptor protein in RNA sensing, is critical for NLRP3 inflammasome activation during infections with RNA viruses and stimulation with the synthetic RNA polyinosinic-polycytidylic acid. MAVS recruits NLRP3, directing its localization to the mitochondria [124]. Still, MAVS is not essential for NLRP3 inflammation induced by other NLRP3 stimuli. Recently, it was shown that trans-Golgi network disassembly into vesicles, known as dispersed trans-Golgi network (dTGN), is another process that leads to NLRP3 inflammasome activation. More specifically, the phospholipid phosphatidylinositol-4-phosphate on dTGN drives NLRP3 aggregation, ASC oligomerization and caspase-1 activation, and downstream signaling [125].

Interestingly, recent studies have implicated cellular metabolic events in the activation of the NLRP3 inflammasome. Indeed, aerobic glycolysis and the mitochondrial electron transport chain (ETC) enhance NLRP3-driven responses [126,127,128,129] (Figure 1). In LPS-stimulated macrophages, activation of mammalian target of rapamycin complex 1 (mTORC1) promotes hexokinase (HK1)-dependent glycolysis which, in turn, induces NLRP3 activation [126,127,128,129]. Consequently, inhibition of mTORC1 or deficiency of Raptor, an mTORC1-binding partner, decreases HK1-dependent glycolysis and suppresses NLRP3 signaling [126,127,128,129]. Moreover, glucose deprivation, 2-deoxyglucose (2-DG) treatment, or HK-1 knockdown suppresses ATP-driven NLRP3 activation and inhibits IL-1β secretion by macrophages [126,127,128,129]. Additionally, saturated fatty acids, such as palmitate, suppress the activation of the anti-inflammatory AMP-activated kinase, leading to increased ROS production and NLRP3 activation [130].

Altogether, these secondary activating signals lead to IL-1β and IL-18 secretion downstream of NLRP3 activation and also to pyroptosis (Figure 1). Still, despite extensive research in understanding the upstream events during NLRP3 activation, there is still no single unifying model, and further studies using genetic approaches, rather than pharmacological inhibition that could lead to indirect and off-target effects, need to be performed.

## 7. Role of the Noncanonical and Alternative Nlrp3 Activation Pathways

The noncanonical pathway of NLRP3 activation is associated with the detection of intracellular LPS generated following infection by Gram-negative bacteria, such as *Escherichia coli*, *Salmonella typhimurium*, *Shigella flexneri*, and *Burkholderia thailandensis* [131,132,133]. As such, the noncanonical NLRP3 pathway induces pyroptotic cell death and restricts the growth of intracellular bacteria in myeloid and nonmyeloid cells. The noncanonical NLRP3 pathway requires signaling through caspase-11 in mice and caspases 4 and 5 in humans. The binding of LPS to caspases 11, 4, and 5 results in their autoactivation and cleavage of GSDMD, triggering pyroptosis and the secretion of IL-1α [134]. Pyroptosis enhances K^+^ efflux which activates the canonical NLRP3 pathway and the release of IL-1β and IL-18. Hence, caspases 4, 5, and 11 do not cleave IL-1β and IL-18, but only lead to pyroptosis, and the subsequent canonical NLRP3 inflammasome activation pathway is responsible for caspase-1 activation and cytokine secretion. Activated caspase-11 also cleaves pannexin-1, a membrane ATP channel, which induces K^+^ efflux and activates the canonical NLRP3 pathway [131,132,133].

The cytosolic accessibility of LPS is driven by Guanylate-Binding Proteins (GBPs) and the Immunity Related GTPase family member 10 (IRGB10) which lyse the Gram-negative bacterium-containing vacuoles, releasing bacterial LPS into the cytoplasm [134,135]. Bacterial outer membrane vesicles (OMVs) also deliver LPS into the cytoplasm through endocytosis. Another process through which extracellular LPS activates the noncononical NLRP3 pathway is the binding and activating of its receptor, TLR4. TLR4 induces TRIF and MyD88 signaling, and drives the production of type I IFNs which, in turn, enhance the expression of the noncanonical inflammasome components, caspase-11, GBPs and IRGB10 [135,136]. In fact, type I IFNs, along with the complement C3-C3aR axis, upregulate caspase-11 expression. In neutrophils, the activation of the noncanonical NLRP3 pathway through detection of cytosolic LPS induces the release of neutrophil extracellular traps (NETs) that, in turn, activate the canonical NLRP3 pathway [137]. Reciprocally, IL-18 released upon NLRP3 inflammasome assembly induces NETosis [137].

An alternative NLRP3 inflammasome pathway was recently discovered in human and porcine monocytes and does not require a secondary signal [106]. LPS recognition by TLR4 induces the intracellular activation of the TIR-domain-containing adapter-inducing interferon-β - Receptor-interacting serine/threonine-protein kinase 1 - Fas-associated protein with death domain - Caspase-8 (TRIF-RIPK1-FADD-CASP8) cascade. Cleavage of caspase-8 induces NLRP3 activation and the maturation of IL-1β and IL-18, through an as yet unknown mechanism. This alternative pathway of NLRP3 activation occurs independently of K^+^ efflux and ASC speck formation. An additional unique feature is that it does not trigger pyroptosis, and the secretion of IL-1β is independent of GSDM [106]. Notably, in mouse bone marrow-derived macrophages (BMDM), simultaneous TLR and NLRP3 stimulation leads to rapid inflammasome activation independent of de novo gene transcription [138]. This type of NLRP3 activation does not promote IL-1β secretion and pyroptosis, but enhances IL-18 production and provides a fast protective response against intracellular pathogen burden. 

## 8. Regulation of Nlrp3 Functions 

The activation of the NLRP3 inflammasome is associated with a diverse range of human diseases. Mutations in the *NLRP3* gene are associated with the dominantly inherited autoinflammatory diseases known as cryopyrin-associated periodic syndromes (CAPS), including familial cold autoinflammatory syndrome, Muckle–Wells syndrome, and chronic infantile neurological cutaneous and articular syndrome [139,140,141]. Single nucleotide polymorphisms in the genes encoding NLRP3 inflammasome components have been also associated with the pathophysiology of Crohn’s disease and rheumatoid arthritis. Other studies have revealed that excessive NLRP3 activation is implicated in diseases driven by metabolic dysfunction such as type 2 diabetes and nonalcoholic steatohepatitis, in neurodegenerative diseases including Alzheimer’s disease and Parkinson’s disease, and in cancer [142,143,144,145]. Notably, NLRP3 activation through exposure to crystals and protein aggregates is associated with silicosis and fibrosis in the lung, atherosclerosis, gout flares, and kidney dysfunction [142]. Hence, stringent regulation of NLRP3 responses is essential for the control of overactive inflammatory processes and the prevention of tissue damage.

NLRP3 regulation takes place both at the transcriptional and the post-transcriptional levels. Type I-IFNs repress the expression of pro-IL-1β through the secretion of IL-10 [146] (Figure 1). Type-I IFNs also induce the expression of iNOS, which inhibits NLRP3 activation through the production of Nitric oxide (NO ) and TRIM that reduces ROS release [147]. Post-transcriptional regulation of NLRP3 involves signaling through the microRNA, mir-223, which binds to the 3’ untranslated region UTR of NLRP3 and inhibits its expression [148] (Figure 1). PTMs can also negatively regulate NLRP3 responses. Indeed, the ubiquitylation of the LLR domain of NLRP3 by the membrane-associated Ring finger protein 7 (MARCH-7), and the phosphorylation of its Ser291 residue, negatively regulates NLRP3 activation [149,150].

One of the most important mechanisms that restrain NLRP3 functions is autophagy (Figure 1). Autophagy is an endogenous recycling process utilized by the host to maintain cell homeostasis in response to stress [151]. In autophagy, dysfunctional or unnecessary cellular components become degraded. In addition, mitophagy promotes the clearance of damaged mitochondria from the cytoplasm and reduces mtROS [151]. Recent studies have shown that activation of NLRP3 in macrophages deficient in autophagy components, such as *Beclin* or *Lc3b*, leads to increased secretion of mtDNA and ROS, and increased activation of caspase-1 and release of IL-1β and IL-18 [152,153]. Furthermore, in vitro treatment of monocytes with rapamycin, an autophagy inducer, reduces IL-1β secretion in response to LPS [154]. In addition, induction of mitophagy through activation of the receptor-interacting serine/threonine-protein kinase 2 (RIP2) limits virus-induced NLRP3 activation [155]. Moreover, infection with influenza A virus in mice deficient in *Nod2* and *Ripk2* results in defective mitophagy, leading to excessive activation of NLRP3 and increased IL-18 production [155].

Signaling through metabolic regulators can also inhibit NLRP3 activation. Dimethyl fumarate (DMF) activates the transcription factor NF-E2-related factor 2 (NRF2) that represses IL-1β and NLRP3 gene expression in LPS-treated microglia and in the human acute monocytic leukemia cell line (THP-1) [156]. NRF2 also regulates levels of antioxidant genes to support cell survival during oxidative stress, and through limiting ROS levels, it inhibits NLRP3 activation [157]. DMF also decreases mtROS release and suppresses NLRP3 assembly [156]. In addition, nicotinamide adenine dinucleotide (NAD^+^) activates sirtuins (Sirt2) which inhibit NLRP3 inflammasome activation and decrease pro-IL-1β production [158] (Figure 1). Some metabolites of fatty acids, including β-hydroxybutyrate and α-linolenic acid, suppress NLRP3 activation via inhibiting K^+^ efflux and the oligomerization of ASC, and by reducing ROS levels [159,160]. Interestingly, cholesterol metabolism is also associated with the regulation of NLRP3 responses. Of note, a recent report documented that bile acids through the TGR5-cAMP-protein kinase A axis inhibit NLRP3 activation and prevent LPS-induced systemic inflammation, alum-mediated peritoneal inflammation, and type 2 diabetes in mouse models [161]. Other studies in human primary monocyte-derived macrophages showed that PGE2 treatment following LPS stimulation inhibits NLRP3 activation through increasing intracellular cAMP levels [162]. In support, inhibition of cyclooxygenase 2, resulting in PGE2 blockade, enhances NLRP3 activation in human macrophages. In response to PGE2, protein kinase A is also upregulated, phosphorylates Ser295 of NLRP3, and attenuates its ATPase function [150]. Conversely, treatment of BMDM with PGE2 prior stimulation with LPS and ATP increased IL-1β and active caspase-1 release in culture supernatants, highligting species-specific differences as well as differences dependent on the timing of PGE2 administration [163].

Several small-molecule compounds have been described as potent inhibitors of NLRP3 activation, including MCC950 [164], β-hydroxybutyrate (BHB) [159], Bay 11-7082 [165], dimethyl sulfoxide (DMSO) [166], and type I IFN [167]. Seminal studies by Coll et al. have documented that MCC950 inhibits both the canonical and the noncanonical pathways of NLRP3 activation, while it does not affect other inflammasomes. MCC950 prevents NLRP3-induced ASC oligomerization without affecting NLRP3 priming, and inhibits IL-1β secretion by human and mouse macrophages [164]. Subsequent studies demonstrated that MCC950 blocks nigericin-induced NLRP3 activation via inhibition of Cl^-^ efflux in LPS-primed BMDMs [168]. Moreover, MCC950 inhibits NLRP3 inflammasome formation through blocking ATP hydrolysis [169]. Of relevance, preclinical studies have shown that MCC950 alleviates the severity of experimental autoimmune encephalomyelitis and prevents neonatal lethality in a model of CAPS [164]. Pharmacological inhibition of NLRP3 with MCC950 also protected against dopaminergic degeneration in a mouse model of Parkinson’s disease (PD), and reduced total leukocytes and inflammatory macrophages in the BAL of mice infected with influenza A virus [170,171].

The aforementioned paragraphs highlight crucial functions for NLRP3 inflammasome activation in the eradication of pathogens, the protection against damaged and/or dying cells, and the maintenance of tissue homeostasis. However, several gaps in our understanding of the precise molecular pathways involved in the initiation and regulation of NLRP3-driven inflammatory responses remain, and represent promising avenues for future research. In the next section, we describe current knowledge on the role of NLRP3 activation in the initiation and propagation of allergic airway inflammation and human asthma, and discuss the implications of excessive NLRP3 responses in the pathogenesis of allergic diseases.

## 9. Role of Nlrp3 Signaling in Allergic Airway Inflammation

NLRP3 inflammasome activation is involved in the initiation and the propagation of allergen-driven inflammatory responses in the airways. Studies using mouse models of allergic asthma induced by adjuvant-free ovalbumin (OVA) and adjuvant (aluminum hydroxide)-coupled OVA, demonstrate enhanced protein levels of NLRP3 and caspase-1, along with elevated IL-1β and TNF-α release by epithelial cells and macrophages in the airways, compared to Phosphate-buffered saline (PBS)-treated mice [172]. Moreover, in mice sensitized with OVA and LPS and challenged with OVA (OVA_LPS_-OVA), as well as in HDM-instilled mice, increased ROS and mtROS production in BAL cells and in primary tracheal epithelial cells was detected, inducing augmented NLRP3, caspase-1, and IL-1β protein expression in the airways [173]. Interestingly, recent studies using three distinct HDM-induced mouse models of allergic airway inflammation (AAI), corresponding to eosiniphilic, mixed granulocytic, and neutrophilic asthma subtypes, documented increased expression of Nlrp3, Nlrc4, Nlrc5, Pycard, Casp-1 genes, and pro-IL-1β protein levels in the lungs, especially in the neutrophilic asthma model, while mature IL-1β was not shown, suggesting that although inflammasome molecules are upregulated, they do not form functional complexes without an additional trigger [174]. Moreover, increasing inflammasome sensor, caspase-1, and pro-IL-1β expression was documented from eosinophilic to neutrophilic asthma, illuminating an association of inflammasome signaling pathways with the type of airway inflammation [174]. Notably, induction of neutrophilic airway inflammation in mice challenged with HDM and polyinosinic-polycytidylic acid increased the concentration of Apolipoprotein E (APOE) in the epithelial lining fluid and enhanced IL-1β levels in the BAL, pointing to a potential role of APOE in IL-1β production [175].

Pertinent to the role of NLRP3 activation in allergic responses, studies using mice deficient in *NLRP3* and *ASC* in a model of OVA-induced AAI demonstrated decreased eosinophil influx, dampened AHR and reduced airway inflammation, and goblet cell accumulation, accompanied by decreased IL-1β expression in the airway, compared to wild type (WT) littermates [176]. Other studies also in a model of OVA-AAI, revealed that *NLRP3^-/-^* mice exhibit decreased pulmonary inflammation with suppressed mucus secretion, Th2 cytokine and chemokine production, and IgE levels [177]. Interestingly, in a model of OVA-AAI, production of IL-1α and IL-1β in the airways propagated Th2 cell-driven allergic responses and exacerbated pulmonary eosinophilia in a process mediated by caspase-8 activation [178]. These studies highlighted a novel role for caspase-8 in NLRP3 activation in the allergic airway, which was independent of caspase-1 and 11 signaling [178]. Other reports using a model of serum amyloid A (SAA)-induced AAI documented that the secretion of IL-1β in response to SAA is dependent on NLRP3 activation [179]. In fact, using mice deficient in *NLRP3* and *caspase-1*, the authors demonstrated a reduction in infiltrating neutrophils in the lung and decreased inflammatory cytokine/chemokine release (IL-1β, IL-6, and MCP-1 or CCL2) in the BAL compared to WT controls [179]. Another study using a complete Freund’s adjuvant (CFA)/HDM-induced mouse model of AAI showed that administration of CRID3, an NLRP3 inhibitor, reduced IL1-β and Th2 cytokine production in the BAL and inhibited AHR [180]. Interestingly, a recent report demonstrated that NLRP3, along with IRF4, transactivates the *Il4* promoter, enhances Th2 cell differentiation, and exacerbates asthma symptoms in a mouse model [181]. Moreover, studies by Kim et al., in an experimental model of high fat diet-induced obesity, demonstrated that obese mice had increased AHR driven by aberrant NLRP3 inflammasome-dependent responses in the adipose tissue, which contributed to the activation of ILCs and increased IL-17 responses in the lung, in the absence of allergic sensitisation [182]. These studies highlighted a novel NLRP3-IL-1β-Th17 link in AHR development in obesity-associated allergic airway disease.

Notably, sensitization with OVA-alum or *Aspergillus fumigates* followed by challenge with OVA or *A. fumigates*, respectively, increased ROS production in lung homogenates mediated by mitochondrial Ca^2+^/calmodium-dependent protein kinase II (CaMK II), which induced NLRP3, active caspase-1 and mature IL-18 at the mRNA, and protein levels in allergic lungs, as well as exacerbated OVA-AAI [183]. In contrast, inhibition of mitochondrial CaMK II reduced AHR, Th2 cytokine production and NLRP3 activation in the lungs [183]. Further studies in a mouse model of OVA-AAI, accompanied by infection with *Chlamydia muridarum* or *Haemofilus influenza*, demonstrated an increased expression of NLRP3 in airway epithelial and infiltrating immune cells, as well as enhanced IL-1β and caspase-1 mRNA levels in the lungs of infected mice [184]. Notably, in vivo administration of MCC950 decreased AHR and neutrophil accumulation in the airways of infected mice [184]. Furthermore, using mice deficient in *NLRP3* and *IL-1β*, another study documented reduced airway inflammation and cytokine release following rhinovirus (RV) infection in HDM-challenged mice, highlighting the implication of NLRP3 activation in RV-induced disease exacerbations [185]. Notably, subepithelial macrophages were the major source of IL-1β in response to RV, with low IL-1β expression by the airway epithelium, suggesting that NLRP3 activation mainly occurred in macrophages. Moreover, no induction of IL-18, either at the mRNA or the protein levels, was observed in airway epithelial cells upon RV infection. It should be emhpasized that, in contrast to the airway epithelium, the gut epithelium is characterized by increased IL-18 and low IL-1β production in response to NLRP3 signaling. Indeed, IL-1β levels remained unaltered in colonic intestinal epithelial cells from mice treated with a high-fibre diet, while active caspase-1 and IL-18 were increased [186]. Notably, using *NLRP3^-/-^* mice, the authors showed that the protective effects of a high-fibre diet against colitis were mediated through NLRP3 activation in colonic epithelial cells. Altogether, these studies suggest that epithelial cells at distinct sites respond differently to inflammasome activation, not only in terms of cytokine secretion but also in the inflammasome sensor utilized, and propose a key role of the tissue micromilieu in governing epithelial cell-induced inflammatory responses. 

Contradictory data have been generated by studies showing that NLRP3 activation is not essential for allergic disease development in OVA and HDM induced AAI. Allen et al. showed that the adjuvant effects of aluminum hydroxide in the OVA-AAI model were not affected in *NLRP3^-/-^*, *Casp^-/-^,* or *PYCARD^-/-^* mice [187]. In another study, *NLRP3^-/-^* mice did not exhibit significant differences in airway eosinophilia, mucus production, AHR, and Th2 cell responses upon exposure to uric acid crystals compared to their WT counterparts [188]. Similar findings were observed in a combined particular matter (PM)/Ova-induced mouse model of experimental asthma [189]. Finally, Madouri et al. demonstrated that mice deficient in *NLRP3^-/-^*, *Casp^-/-^,* or *PYCARD^-/-^* exhibited enhanced lung inflammation and pathology, including eosinophilic inflitration and Th2 cytokine release, upon exposure to HDM, supporting the notion that NLRP3 activation exerts protective functions against HDM-induced allergic lung inflammation [190].

## 10. Nlrp3 Signaling in Human Asthma

A series of recent studies suggest that NLRP3 activation is involved in human asthma pathogenesis. Hirota et al. was among the first to describe NLRP3 and caspase-1 protein expression in human lung sections and in primary airway epithelial cells from healthy volunteers following exposure to PM_10_. In fact, they demonstrated that NLRP3 silencing using short hairpin (sh) NLRP3 attenuated PM_10_-induced release of IL-1β by human airway epithelial cells [191]. In other studies, in vitro RV infection of HBECs upregulated NLRP3, NLRC5, and caspase-1 protein levels that triggered IL-1β secretion [192]. Knock down of NLRP3 or NLRC5, using shRNA, decreased IL-1β secretion by HBECs, while simultaneous knock down of NLRP3 and NLRC5 abrogated IL-1β secretion. Moreover, HBEC from asthmatics exhibited enhanced co-localization of caspase-1 and ASC and increased mRNA expression of caspase-4 after IAV infection compared to healthy controls [193] (Figure 2). Still, it should be emphasized that the previous studies were using human bronchial epithelial cells cultured in a monolayer, and NLRP3 activation in airway epithelial cells cultured in an air–liquid interface (ALI), which is a more physiologically relevant model, remains incompletely defined. Increased NLRP3 and IL-18 protein levels were observed in airway epithelial cells in lung biopsies from asthmatics compared to healthy individuals [183]. Notably, individuals with neutrophilic asthma had elevated mRNA levels of NLRP3, caspase-1, and IL-1β, as well as NLRP3 and caspase-1 protein expression in sputum macrophages and neutrophils, compared to eosinophilic and paucigranulocytic asthmatics [180,194] (Figure 2). Increased expression of NLRP3 and IL-1β was also detected in the sputum of patients with SA compared to MMA, and correlated with clinical parameters of disease, such as neutrophilic airway inflammation, Asthma Control Questionnaire (ACQ) score, and Forced Expiratory Volume in 1 second (FEV1)% [184]. In another study, increased extracellular DNA (eDNA) sputum levels in SA correlated with sputum neutrophilic inflammation, increased NETs formation, caspase-1 activity, and IL-1β levels [195]. NLRP3 gene expression and IL-1β protein levels were also increased in sputum inflammatory cells from obese asthmatics compared to non-obese asthmatics, and correlated with body mass index [196] (Figure 2). Kim et al. showed higher expression of NLRP3 and caspase-1 in the BAL of asthmatics compared to healthy subjects [173] (Figure 2). BAL macrophages from asthmatics treated ex vivo with HDM also upregulated *NLRP3* and pro-IL-1β expression, resulting in increased IL-1β secretion in an APOE-dependent manner [174]. Lui et al. also showed that macrophages isolated from the PB of patients with Th2/Th17-predominant asthma had higher mRNA and protein levels of NLRP3 components and IL-1β compared to healthy controls [197].

Pertinent to inflammasome related-cytokines, most studies have shown that IL-1β is increased in the sputum and BAL of patients with neutrophilic asthma [198], and in the serum of asthmatic patients with or without steroid treatment, compared to controls [199] (Figure 2). Contradictory findings were observed regarding IL-18 release, with some studies showing increased IL-18 in the serum [200,201] and sputum from SA patients [202,203], and others reporting decreased levels in induced sputum [204]. Considering the increased activation of the NLRP3 pathways, along with excessive IL-1β release, in asthmatic patients, the concept of IL-1β blocking as a therapeutic approach in SA appears promising. Therapeutic administration of canakinumab, a fully human anti-IL-1β monocloncal antibody, has been widely used in conditions ranging from CAPS to rheumatoid arthritis, atherosclerosis, and lung cancer [205,206,207,208]. Pertinent to asthma, there was only one randomized double-blind placebo-controlled study that evaluated the safety and tolerability of canakinumab in mild asthmatic patients, as well as its effects on the attenuation of the late asthmatic response (LAR) following allergen challenge [209]. Canakinumab appeared to be safe and attenuated LAR compared to pretreatment values. Despite these positive results, no further studies have been conducted since [209], and canakinumab is no longer under investigation as a treatment for asthma (searched on clinicaltrials.org on 11/09/2019).

In summary, a growing body of evidence suggests that NLRP3-induced inflammasome responses are implicated in AAI both in experimental models and human asthma (Figure 2). Still, certain controversial results obtained in animal studies are mainly associated with variations in the experimental models utilized, including type and concentration of allergen, route and time of administration, as well as mouse strain differences. The observed differences could be also associated with the distinct housing conditions affecting the microbiota composition. Hence, further mechanistic studies are warranted to resolve these disparities. More importantly, the precise role of NLRP3 activation in experimental models of SA remains elusive and deserves investigation. Of note, certain findings observed in animal studies were not observed in human asthmatics. For example, in human airway epithelial cells, it seems that other than NLRP3 inflammasome sensors become activated and trigger IL-1β release. Moreover, the precise factors that initate inflammasome assembly and activation in asthmatics remain incompletely defined. Mechanistic studies using animal models that more closely resemble SA need to be performed to clarify the type of cells in which NLRP3 signaling is activated, as well as its upstream regulators. In addition, functional studies using human primary airway epithelial cells in ALI cultures are essential for the elucidation of the differences in NLRP3-induced signaling in humans and mice. Finally, elucidation of the role of NLRP3 activation in the functional crosstalk between airway epithelial cells and other lung structural cells, such as ASMs, may shed new light on the mechanisms underlying tissue remodelling in SA.

## 11. Concluding Remarks

The past decade has witnessed a burgeoning appreciation of the existence of a wide range of SA endotypes. Still, and particularly in the type 2 low asthma endotypes, there is a considerable gap in our understanding of the cellular and molecular mechanisms involved and a remarkable scarcity of relevant biomarkers. More importantly, no effective treatments targeted at these endotypes have emerged. Growing evidence has illuminated a key role for NLRP3 inflammasome activation in the development and exacerbation of allergic responses. Hence, targeting NLRP3 inflammasome pathways in the airways of allergen-challenged mice, particularly in the context of SA, will improve our understanding of how NLRP3 signaling contributes to the development of specific aspects of disease severity. In fact, the direct comparison of the expression and activation of NLRP3 in mouse models and patients with distinct asthma severities is essential for the identification of novel biomarkers pertinent to the diverse asthma endotypes.

Current treatment modalities of NLRP3-related inflammatory human diseases target IL-1β with IL-1β antibodies or recombinant IL-1βR antagonists, such as canakinumab and anakinra, respectively. Nevertheless, most of these inhibitors are relatively nonspecific and have low efficacy. Thus, the development of targeted NLRP3 inflammasome site-specific therapeutics may be more beneficial in suppressing inflammasome-associated disease whilst not predisposing to infection. However, a deeper understanding of the NLRP3 inflammasome assembly and activation is needed before translation of these findings into therapies in clinical practice, especially in the context of SA. To achieve this, we need the development and use of better in vivo models of SA, along with complementary human studies using physiologically relevant in vitro models. Ultimately, this will facilitate the development of personalized medicine for the growing numbers of patients with SA.

## Figures and Tables

**Figure 1 jcm-08-01615-f001:**
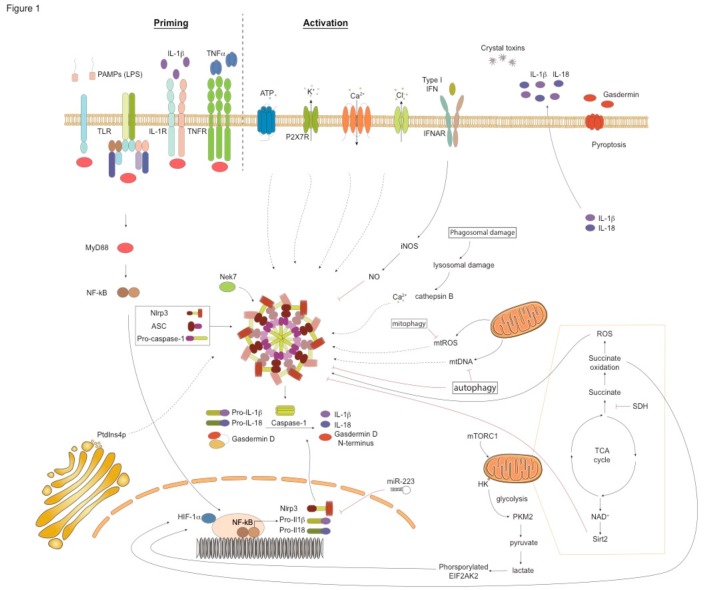
**Mechanisms involved in activation and regulation of NLRP3 canonical pathway.** NLRP3 inflammasome activation requires two signals. The “priming” signal is triggered by PAMP/DAMP recognition by PPRs (e.g. TLRs) and certain cytokines (e.g. TNF-α, IL-1β) and activates NF-κB in the cell nucleus. This leads to NLRP3, pro-IL1-β and pro-IL-18 gene transcription. The second signal induces the assembly of NLRP3, ASC, and caspase-1 to form an active NLRP3 inflammasome and ultimately leads to the release of mature IL-1β and IL-18. Gasdermin D is cleaved and becomes inserted into the cell membrane, forming pores and inducing pyroptosis. The mechanisms proposed for the second NLRP3 activating signals are shown and include: a) changes in cytosolic levels of ions, such as K^+^, Cl^-^ and Ca^+2^, b) lysosomal destabilization and the release of cathepsins, c) mitochondrial dysfunction-derived signals such as mtROS, mtDNA and d) metabolic changes. PtdIns4P on dTGN drive NLRP3 activation. Aerobic glycolysis pathways and the TCA cycle also activate NLRP3. Autophagy and mitophagy inhibit NLRP3 inflammasome activation. IFNs also inhibit NLRP3 activation through NO production. IL-1R, IL-1β receptor; TLR, Toll-like receptor; TNFR, tumor necrosis factor receptor; IFNAR, IFNα/β receptor; NEK7, NIMA- related kinase 7; NF- κB, nuclear factor- κB; P2X7, P2X purinoceptor 7; PtdIns4P, phosphatidylinositol-4-phosphate; PYD, pyrin domain; ROS, reactive oxygen species; HK1, hexokinase; mTORC1, rapamycin complex 1; SDH, succinate dehydrogenase; EIF2AK2, eukaryotic translation initiation factor 2-alpha kinase 2.

**Figure 2 jcm-08-01615-f002:**
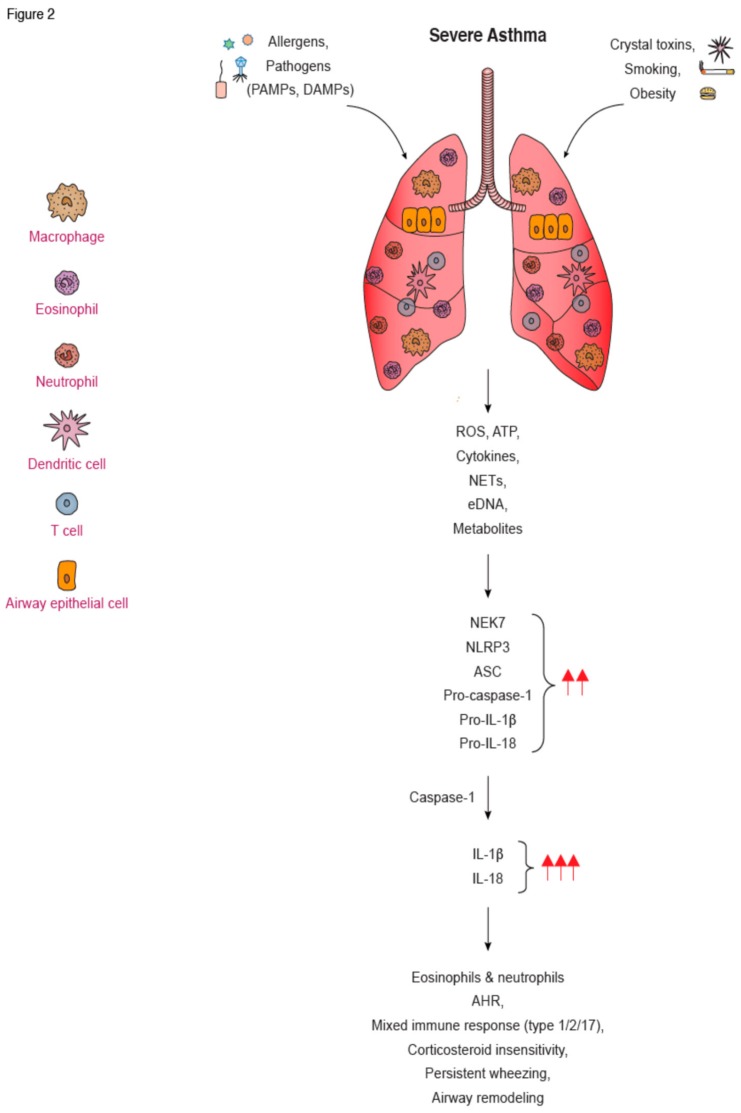
**The role of NLRP3 inflammasome in the development of severe asthma**. Exposure to pathogens, allergens, cigarette smoke, and other noxious stimuli in the asthmatic airway triggers the production of ROS, cytokines, and NETs which, in turn, can activate the NLRP3 inflammasome in infiltrating eosinophils, neutrophils, and macrophages, as well as in airway epithelial cells. This results in the enhanced release of IL-1β and IL-18, which leads to increased Th1 Th2 and/or Th17 cell infiltration and associated pathological consequences, such as mucus hypersecretion, AHR, and airway remodelin. eDNA, extracellular DNA; NETs, neutrophil extracellular traps; AHR, airway hyperresponsiveness.

**Table 1 jcm-08-01615-t001:** Key characteristics of canonical, noncanonical and alternative NLRP3 activation.

Pathways of NLRP3 Inflammasome Activation	Canonical	Noncanonical	Alternative
**NLRP3-ASC-Casp1 signaling**	Yes	Yes	Yes
**Caspase-1 cleavage, IL-1β maturation**	Yes	Yes	Yes
**Caspase-11, 4, 5 cleavage**	No	Yes	No
**Caspase-8 cleavage**	No	No	Yes
**Signal 2 requirement**	Yes	No	No
**Pyroptosome formation**	Yes	Yes	No
**Cell Death**	Yes	Yes	No

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
