# Peer review of "Targeting NLRP3 Inflammasome Activation in Severe Asthma"

_jcm, 2019, doi:10.3390/jcm8101615_

Round 1
Reviewer 1 Report
The current review manuscript efficiently summarizes the recent literature on field of asthma and NLRP3 inflammsome. The literature review has been extensive and well presented. There are few minor points to be taken into account and revise prior to final submission;
1- Page 1 : "exposure, respiratory infections exercise, climate changes and stress"; a comma missing between infections and exercise
2- In chapter "Type 2 asthma"; it would be better of the authors refer to Th9 subset and its implications in pathogenesis of asthma with some reference.
3- Page 5; There is no mention of potential/reported adverse effects of administration of monoclonal antibodies (against IgE, IL-5, IL-5R, ...) in asthmatic patients.
4-Page 9: "of mtDNA, which activates NLRP3 inflammasom. Increased mtROS production oxidazes" Typos, correct as follows: Inflammasome \ oxidizes.
MAVs: Change to MAVS
5- Page 10: "the secretion of IL-1a and HMGB-1" : Change IL-1a to IL-1α
6- Page 11: "Notably, in mouse bone marrow-derived macrophages simultaneous TLR and NLRP3 stimulation, through TLR2, TLR4, TLR7 or TLR9, together with extracellular ATP, leads to rapid inflammasome activation independent of de novo gene transcription. This type of NLRP3 activation does not ..." : Provide reference for this paragraph.
7- Page 12: "and prevent metabolic disorders, such as, LPS-induced systemic inflammation, alum-induced peritoneal inflammation and type-2 diabetes" :
It is not clear why the authors calssify LPS-induced systemic inflammation, alum-induced peritoneal inflammation as "metabolic disorders" which they are not.
8- Page 13: "and decreased inflammatory cytokine release (IL-1β, IL-6 and MCP-1 or CCL2)" : correct to: "cytokine/chemokine"
9- Figure 2: In the text authors have discussed implication of roles of NLRP3 inflammasome components in airway epithelial cells, yet there is no mention of this cells in the figure. Please add the epithelial cells to the figure/caption.
10- Page 15: Any reports of IL-1 blockade in asthma clinical trials ? Please provide references, if any.
11- Change personalised to personalized.
Reviewer 2 Report
In the manuscript entitled “Targeting NLRP3 inflammasome activation in severe asthma" the Authors provide a comprehensive review on severe asthma (SA) characteristic, pathomechanisms, epidemiology and treatment therapies, combined with broad and up-to-date description of NLRP3 inflammasome activation mechanism and its link to allergic diseases and SA. Submitted review covers broadly investigated, significant and interesting field of research with potent impact on available treatment options. To assure the highest quality of the submitted paper there are several suggestions to be considered:
Major points
To allow an in-depth understanding of all the information included in the manuscript supplementing the references with original papers, not only the reviews, would be necessary. For better comprehension of pathogenesis of SA paucigranulocytic asthma description should be added to the paper. The good overview of this inflammatory phenotype of asthma can be found here: Tliba O et al. J Allergy Clin Immunol. 2019 Apr;143(4):1287-1294. To provide a better overview of NLRP3 inflammasome control mechanisms, MCC950 inhibitor should be described in more details, starting from Coll RC et al Nat Med. 2015 Mar;21(3):248-55. Anti-IL1ß biological treatment (canakinumab) is widely and successfully tested in different disorders (Ridker, P.M., et al. N Engl J Med 2017; 377:1119-1131; Lachmann, H. J., et al. N Engl J Med 2009; 360:2416-2425).Potential therapeutic targeting of IL-1bin the context of NLRP3 inflammasome activation in SA should be discussed. The fragments about the expression of NLRP3 and activation of NLRP3 inflammasomein bronchial epithelial cells should be carefully rewritten. It would be good to add the distinction between NLRP3 gene and protein expression in mouse and in human bronchial epithelium and studies showing clearly full NLRP3 activation. In fact, to date there are no studies showing convincingly activation of NLRP3 inflammasome in differentiated human epithelium (and limited in mice). It would be good to add that the studies showing NLRP3 activation in vitro were using bronchial epithelial cells in monolayer and not in air-liquid interphase (ALI) (which is more physiological model of tracheal and bronchial epithelial cells). Those, which were using ALI cultures showed activation of inflammasome and production of mature IL-1ß, but not exactly via NLRP3. Many studies are showing that NLRP3 activation happens in airway macrophages and dendritic cells, whereas in epithelium other inflammasome sensors might be responsible for production of mature IL-1ß in inflammasome-dependent manner. Definitely NLRP3 inflammasome activation has been convincingly shown in intestinal epithelium, but the issue of airway epithelium should be clarified. A recently published paper describing a novel model of three inflammatory phenotypes (eosinophilic, mixed and neutrophilic) of allergic airway inflammation (AAI) in mice confirms that neutrophilic AAI shows inflammasome fingerprints, which is further confirmed with human samples. The manuscript provides in vivo model for the in-depth study of distinct mechanisms related to different inflammatory phenotypes of asthma. Due to its relevant impact on the field, it should be discussed in the paper (Tan, H-T. T., et al. Allergy. 2019;74:294–307). It is broadly discussed, that inflammasome activation in the lung is characterized rather by IL-1ß secretion, whereas gut shows more pronounced IL-18 expression profile (Minguyan, H., et al. Mucosal Immunology,12, 958–968 (2019); Macia, L., et al. Nature Communications, 6: 6734 (2015). For a complex understanding of inflammasome activation in SA, this issue should be included and discussed in details. In the chapter: “NLRP3 biology and function” ASC specs internalization by macrophages is discussed. Importantly, detection of ASC specks in the human plasma in the in vivo model has been reported (Basiorka, A. A., et al, Blood 2016 128:2005). It would be beneficiary to review the local inflammasome signature in the context of potential activation of distant cell populations and systemic inflammation. The review discusses the influence of fatty acids on NLRP3 inflammasome activation. Prostaglandin E2(PGE2), a derivative of arachidonic acid, one of the most common n-3 fatty acid, is an important lipid mediator with a complex influence on asthma. In-depth mechanistic research showed that PGE2 bear the potential to inhibit NLRP3 inflammasome activation (Sokolowska, M., et al. J Immunol.2015 Jun 1;194(11):5472-5487). It was later confirmed by others Mortimer L., et al. Nat Immunol. 2016 Oct;17(10):1176-86.). Discussing the effect of eicosanoids in the context of the review topic would add to the quality of the paper. It is also interesting in the context of differential regulation and eicosanoids release in severe asthma (Sokolowska M. et al. J Allergy Clin Immunol. 2017 Apr;139(4):1379-1383.)
Minor comments
Introduction: In the sentence "Factors, such as allergen or irritant exposure, respiratory infections,exercise, climate changes, and stress are responsible for the disparities and severity of asthma symptoms" comma is missing between respiratory infections and exercise part of paraphrase. In the part: Inflammasomes: a key component of innate immunity in the sentence “PRRs consist of the Toll-like receptors (TLRs), the RIG-I receptors (RLRs), the nucleotide-binding oligomerization domain-like receptors (NLRs), the Scavenger receptors, the C-type lectin receptors (CLRs) and the absent-in-melanoma (AIM)-like receptors (ALRs)” part “Rig like receptors” should be corrected to “RIG-I-like receptors”. In the chapter: Role of NLRP3 signaling in allergic airway inflammation, in the sentence “Taken together and considering the key role respiratory tract infections play in the development of SA, there studies suggest, that infection-induced, NLRP3 inflammasome-mediated IL-1b responses are critical for disease development” typing mistake in the word “taken” should be corrected.
Reviewer 3 Report
This review summaries numerous studies related biology and functions of NLRP3 and the role of NLRP3 in the pathogenesis severe asthma. The review is overwhelmed with information about large network of pathways leading to severe asthma symptoms. The review covers extensive literature, but however it can be improved by reorganizing the presentation to emphasizing the most important aspects of the key pathways, and the other, likely less important features reduce, so the clear message is presented to broader audience. In present form it is relatively easy to be lost in a web of extensive information.
Also, it will beneficial to include a paragraph related to translation of findings in mice and d other species to humans. Critical view what could be the differences between the experiments in different species and under different, frequently unphysiological conditions. Are the pathways the same? Is the flow along the pathways the same?
Round 2
Reviewer 2 Report
The authors of the review entitled “Targeting NLRP3 inflammasome activation in severe asthma” thoroughly improved manuscript, carefully taking into consideration the reviewers’ comments. There are two minor suggestions, that could be taken into consideration before publishing:
In the review authors discuss not yet entirely clear role of NLRP3 in severe asthma. In light of some contradictory reports, a statement in the description of the Figure 2: “Exposure to pathogens, allergens, cigarette smoke and other noxious stimuli in the asthmatic airway, triggers the production of ROS, cytokines and NETs, which, in turn, activate NLRP3 inflammasome in infiltrating eosinophils, neutrophils and macrophages, as well as in airway epithelial cells” should be rather more general e.g. “Exposure to pathogens, allergens, cigarette smoke and other noxious stimuli in the asthmatic airway, triggers the production of ROS, cytokines and NETs which, in turn, can activate NLRP3 inflammasome.” Line 591-596. As far, as this reviewer is aware, it would be to better to conclude: “Interestingly, recent studies using three distinct HDM-induced mouse models of allergic airway inflammation (AAI), corresponding to eosinophilic, mixed granulocytic and neutrophilic asthma subtypes, documented increased expression of Nlrp3, Nlrc4, Nlrc5, Pycard, Casp-1 genes and pro-IL‐1β protein levels in the lungs, especially in the neutrophilic asthma model, while mature IL‐1β was not shown, suggesting that although inflammasome molecules are upregulated they might not form functional complexes without additional trigger.”Author Response
"Please see the attachment."
